🔓 | **Open Peer Review** | Bacteriology | Methods and Protocols

# Generating gnotobiotic bivalves: a new method on Manila clam (*Ruditapes philippinarum*)

Marialaura Gallo,[1] Andrea Quagliariello,[1] Giulia Dalla Rovere,[1] Federica Maietti,[1] Barbara Cardazzo,[1] Luca Peruzza,[1] Luca Bargelloni,[1] Maria Elena Martino[1]

**ABSTRACT** Understanding how microbiomes influence the physiology of animal hosts requires detailed mechanistic insights, often obtained through gnotobiological approaches. Model organisms are central to this research, offering key advantages such as experimental tractability, reproducibility, and ease of manipulation. However, there is a lack of established gnotobiotic models for the marine environment—especially for bivalves, which play a critical role in ecosystem functioning. This gap is particularly important in the context of climate change, where harnessing microbiome resilience could mitigate environmental challenges and enhance host responses. In this study, we present a protocol to generate microbiome-depleted and gnotobiotic clams (*Ruditapes philippinarum*), one of the most widely farmed molluscs in the world and a key sentinel species for environmental and climate change impacts. Our microbiome depletion protocol effectively eliminated all detectable bacterial genera in the clams, with the exception of *Endozoicomonas elysicola*, which was identified solely by 16S rRNA amplicon sequencing and not by cultivation methods. In addition, we developed a microbiome transplantation protocol using inoculation of a mock bacterial community that successfully colonized the recipient clams within 1 h of transplantation. By extending gnotobiotic methods to marine invertebrates, this work opens new avenues for investigating microbial influences on ecologically and economically important species, particularly under the pressure of a changing climate.

**IMPORTANCE** The extensive diversity of host-microbe symbioses across ecosystems requires the use of different models to identify conserved and specific processes underlying such relationships. The need for novel models is particularly relevant in the context of the rapid environmental modifications due to climate change. Bivalve molluscs play a crucial role in the functioning of marine ecosystems. In this study, we present the first experimental protocol for the generation of gnotobiotic clams of the species *Ruditapes philippinarum*, one of the most widely farmed molluscs in the world, and a sentinel organism for environmental pollution. Our work extends the current technical understanding of the establishment of gnotobiotic animals, providing an important method for testing research hypotheses on a key taxonomic group in animal ecology. This study will also open new avenues for investigating the influence of microorganisms on animal health and elucidate the transferability of mechanisms studied predominantly in vertebrates to marine invertebrates.

**KEYWORDS** microbiome, gnotobiology, clams, Ruditapes philippinarum, germ-free organism

T he microbial communities associated with animal hosts, collectively known as the microbiome, are widely recognized for their significant influence on host physiology and the ecology of entire ecosystems (1, 2). As a result, understanding the composition

**Peer Reviewer** Zhen Zhang, Feed Research Institute, Chinese Academy of Agricultural Sciences, Beijing, China

Address correspondence to Luca Bargelloni, luca.bargelloni@unipd.it, or Maria Elena Martino, mariaelena.martino@unipd.it.

The authors declare no conflict of interest.

See the funding table on p. 12.

and functions of animal microbiomes has received considerable attention over the past two decades. The ease of access to genomic data from many organisms, especially microbes, thanks to the rapid development of sequencing techniques, has resulted in a wealth of descriptive information, facilitating the understanding of microbiome composition and identity in different host environments, from plants and animals to broader ecosystems.

In this context, research on host-microbiome interactions has made extensive use of such data, correlating microbial composition with a variety of health and disease conditions (3–5). However, the ease of data collection presents a significant risk of obtaining information without a clear understanding of how to interpret it. This, combined with the challenges of assessing the functionalities of microbiomes due to the complex ecological factors influencing them, can often lead to an overemphasis on their role in animal health (6). Identifying the mechanisms by which microbes shape specific host responses requires starting with a thorough description of microbial identity, ideally down to species or even strain level, and using carefully designed experimental methods aimed at establishing a causal relationship. In this context, the long-standing use of gnotobiotic animals (i.e., animals in which normal host microbiota has been replaced by a defined set of microbes) has provided invaluable tools and insights. The generation of germ-free (GF) animals, ranging from guinea pigs to chickens, goats, and a variety of other mammals, birds, and amphibians (7–11), along with the creation of model organisms (i.e., *Drosophila melanogaster* [12], zebrafish [13], laboratory mice [14]) harboring defined microbial communities, thanks to the development of microbiome transplant protocols, marked a revolutionary advance in host-microbiome research. This breakthrough has facilitated the testing of diverse hypotheses in different ecological contexts, with the aim of unraveling the intricate mechanisms by which microbes affect our lives.

Model organisms offer significant advantages, such as the high reproducibility of experiments across laboratories and the ease with which a variety of different ecological and molecular mechanisms can be dissected (15, 16). However, they represent only a limited part of animal diversity and provide little or no information on species of major ecological or economic importance (17). In particular, the increasingly recognized importance of marine ecosystems calls for specific models to understand how host-associated microbial communities influence the ecology and evolution of marine animal hosts. The need for novel models is particularly relevant in the context of the rapid environmental modifications due to climate change. It is predicted that the effects of climate change will be particularly significant for coastal ecosystems, which are among the most productive and richest (in terms of biodiversity) habitats. Although most gnotobiotic research in aquatic animals has been conducted in zebrafish (13, 18), successful efforts have extended to other species, such as the platyfish (*Xiphophorus maculatus*) (19), tilapia (*Tilapia macrocephala*) (11), medaka (*Oryzias latipes*) (20), rainbow trout (*Oncorhynchus mykiss*) (21), several salmonid species (22), the sheepshead minnow (*Cyprinodon variegatus*) (23), and the starlet sea anemone (*Nematostella vectensis*) (24). Aside from a few studies on amphibians (25) and some coral species (26, 27), however, gnotobiotic research has predominantly focused on vertebrate species. Bivalve molluscs (e.g., clams, mussels, and oysters) provide a crucial role in marine ecosystem functioning. They act as filter feeders, actively filtering water and particulates and creating substrates that serve as habitats for many other species (28). One of the most recognized ecosystem services provided by bivalves is nutrient remediation. By filtering phytoplankton and accumulating nitrogen and phosphorus, they effectively absorb excess nutrients from the environment, including those from human activities such as agriculture and aquaculture (29). Bivalves have also been proposed to act as carbon sinks or sources, but their exact contribution in this respect remains unclear. Because of their feeding behavior and their limited mobility, bivalves have been successfully used as bio-indicators of environmental quality (30). Finally, extractive species, including bivalves, currently account for around half of all aquaculture production and have the

potential to contribute significantly to the sustainable growth of the global aquatic food supply although bivalve aquaculture will be the most affected by climate change (31).

Due to their ecological and economic importance, a gnotobiotic model for bivalves would, therefore, be essential to understand how host-associated microbial communities influence their response to environmental stressors and provide an experimental system to test hypotheses generated by the wealth of microbiome sequence data available for this taxonomic group.

In this study, we present a method for generating germ-free and gnotobiotic clams of the species *Ruditapes philippinarum*, the Manila clam. This is one of the most widely farmed molluscs in the world, it has been used in several studies on the effects of environmental pollution, and its response to climate change has been assessed through an integrative biological approach (32).

The protocol presented here extends the current technical understanding of the establishment of gnotobiotic animals, providing the first model for a key taxonomic group in the marine realm. This will open new avenues for investigating the influence of microorganisms on animal health and elucidate the transferability of mechanisms predominantly studied in vertebrates to marine invertebrates.

## MATERIALS AND METHODS

### Clam maintenance and acclimation

Clams (average shell length 22.5 ± 1.9 mm, average soft tissue wet weight 1.0 g ± 0.2 g) were purchased from the SATMAR hatchery (France). A total of 90 clams were placed in a 20 L aquarium containing artificial seawater (ASW, Aquaforest Sea Salt) at 33 PSU salinity and kept at room temperature. The water used has been purchased as osmotic water with low fixed residue (<30 mg/L) and no heavy metals. Adequate water oxygenation was granted by airstones that continuously bubbled air in the tanks, ensuring no hypoxic/anoxic zone could develop in the tank. After 1 h, nine clams (three per sample, three samples in total, T0, Table 1) were sacrificed for microbial load and 16S rRNA analysis. The aquarium was then placed in an incubator at an initial temperature of 18°C. To avoid acute thermal shock, the clams were acclimated for 5 days, during which the temperature was slowly increased up to 25°C. The final acclimation temperature (25°C) reflected the temperature of the subsequent experimental phases. The animals were fed once per day with New Coral Fito Concentrate (A.G.P., Italy), a commercial mixture of microalgae composed of Isochrysis (T-Iso) (33.3%) + Nannochloropsis (31%) + Tetraselmis (18%) + Phaeodactylum (18%), at a final concentration of ~40 × 10^6 cells/L. To assess bacterial abundance, the food was plated on rich microbiological media (e.g., Marine Agar (MA) and Luria Bertani (LB) agar, Condalab, Spain), and the population size was found to be <10 CFU/mL (data not shown). The commercial diet was always aliquoted under a microbiological hood and stored at +4°C. Water changes were made every 48 h to prevent the accumulation of toxic compounds (e.g., nitrites and ammonia). Approximately 50% of the aquarium water was replaced with fresh ASW at the appropriate temperature.

### Antibiotic treatment for microbial depletion in clams

#### Physical space considerations for optimal sterility

Prior to antibiotic treatment: (i) all items, such as air stones and water pumps, were cleaned with 70% ethanol and placed under UV light for 20 min; (ii) surfaces and equipment, such as incubators and aquaria, were cleaned with 70% ethanol; (iii) the artificial seawater was plated on rich microbiological media such as MA and LB agar (Condalab, Spain) to check its sterility; and (iv) all aquaria were kept in closed incubators throughout the antibiotic treatment period.

**TABLE 1** The time points of clam sampling and experimental conditions (ATB, antibiotic)

| Time | Description |
|------|-------------|
| T0 | Received from hatchery |
| T1 | 5 days post-acclimation |
| T2 | 6 h post-ATB administration |
| T3 | 20 h post-ATB administration |
| T4 | 1 h post-transplant |
| T5 | 6 h post-transplant |
| T6 | 22 h post-transplant |

### Antibiotic treatment

After 5 days of acclimation, nine clams (three per sample, three samples in total, T1, Table 1) were sacrificed for microbial load and 16S rRNA analysis. The remaining clams ($n = 60$) were divided into four aquaria. Each aquarium was set up with 15 clams and 3 L of ASW and kept at 25°C. Three aquaria were used for antibiotic treatment, and the remaining one was used as a control group (i.e., no antibiotic treatment) and placed in a separate incubator. Treated groups received a mixture of five antibiotics at two times: T1, i.e., at the end of the acclimation period and T2, i.e., 6 h after the first antibiotics administration. The set of antibiotics and corresponding concentrations and timing of inoculation were chosen to maximize bacterial perturbation and were validated in preliminary experiments (data not shown). Specifically, antibiotic efficacy was evaluated using the disc diffusion method on MA plates. Eight antibiotics were initially screened to identify the most effective candidates: (i) erythromycin (15 µg), (ii) ampicillin (10 µg), (iii) streptomycin sulfate (10 µg), (iv) ciprofloxacin (5 µg), (v) cefotaxime sodium (30 µg), (vi) kanamycin (30 µg), (vii) penicillin (10 µg), and (viii) tetracycline (30 µg). All antibiotics were supplied by Oxoid Limited (UK). For each antibiotic, three discs were placed on individual MA plates to ensure uniform spacing and to avoid overlapping zones of inhibition. Antibiotic efficacy was tested against two bacterial sources: (1) whole clam homogenates and (2) enriched bacterial cultures derived from clam homogenates. Each inoculum was streaked separately onto the plates containing the antibiotic discs, followed by incubation at 22°C for 48 h. Antibiotics with clear inhibition zones (i.e., erythromycin, streptomycin, ampicillin, cefotaxime, and ciprofloxacin) were selected for subsequent combination treatments in aquarium experiments. Erythromycin and streptomycin act by inhibiting bacterial protein synthesis (33, 34), ampicillin and cefotaxime target bacterial cell wall synthesis (35), while ciprofloxacin targets nucleic acid synthesis (36). The antibiotics were administered by pipetting the mixed solutions into the treatment tanks. To increase the uptake of the antibiotics, the doses were administered after the clams had been fed. The antibiotic mixture consisted of erythromycin (83 mg/L dissolved in 16 mL of EtOH 20%; Vol = 48 mL/aquarium), ampicillin (83 mg/L dissolved in 2 mL of EtOH 20%; Vol = 6 mL/aquarium), streptomycin sulfate (20 mg/L dissolved in 0.8 mL of sterile water; Vol = 2.4 mL/aquarium), ciprofloxacin (20 mg/L dissolved in 20 mL of EtOH 20%; Vol = 60 mL/aquarium), and cefotaxime sodium (20 mg/L dissolved in 0.4 mL of sterile water; Vol = 1.2 mL/aquarium). The control group received an equivalent volume of Milli-Q water and 20% ethanol. At T2 and T3 (6 h and 20 h after antibiotic administration, respectively, Table 1), three clams from each aquarium were collected, sacrificed, and pooled (one sample per aquarium) for the assessment of microbial load and 16S rRNA analysis.

### Microbiome transplant

#### Selection of the mock community

The bacterial strains used to create the mock community for the microbiome transplant were isolated from clam homogenates. The aim was to select bacteria that, as natural members of the clam's microbiota, would effectively colonize the host during the

transplantation process. To identify candidate strains, six clams from the acclimation aquarium were homogenized together for 2 min at 400 rpm (maximum speed) using a Stomacher 3500 (VWR, Italy). After homogenization, 100 µL of the filtrate was serially diluted and plated on different selective and differential media, including MA, iron agar, and thiosulfate citrate bile sucrose (TCBS) agar (Condalab, Spain). Plates were incubated at 22°C for 48–72 h. From the resulting growth, three morphologically distinct bacterial colonies were selected to facilitate detection in terms of microbiological plating after transplantation. Part of each selected colony was used for DNA extraction, while the remaining colonies were subcultured on MA at 22°C for a further 48–72 h. After incubation, the cultures were stored in 80% glycerol at −80°C for future use.

For DNA extraction, the colony was dissolved in 100 µL Milli-Q water and boiled at 95°C for 10 min. After centrifugation at 10,000 rpm for 5 min, the pellet was discarded and the DNA-containing supernatant was immediately used for 16S rRNA gene amplification (see section below for details on 16S rRNA amplification) as described previously. The resulting PCR products were visualized by 1.5% agarose gel electrophoresis and subsequently purified using the ExoSAP PCR Product Cleanup Kit (Applied Biosystems, USA). Sanger sequencing of the 16S rRNA gene was performed at BMR Genomics Company, Italy. Species identification of each colony was achieved by comparing the partial 16S rRNA gene sequences obtained (100–450 nt) with those in the GeneBank database, using web-based BLAST software (37).

### Transplant of the mock community

For the transplant procedures, a total of 90 clams purchased from the SATMAR hatchery (France) were acclimated under the same conditions as described above. After acclimation, clams were divided into four aquaria (15 clams per aquarium). Germ-free (GF) clams were obtained for all aquaria as described above. Twenty hours after antibiotic administration (T3, Table 1), GF clams were transferred to small aquaria containing 1.5 L of ASW for the 2 h depuration phase. Subsequently, 12 GF clams from individual tanks were transferred to new tanks containing 450 mL of ASW. At this point, the clams were fed once, and no further feeding occurred during the transplantation phase. To prepare the bacterial suspension for microbiome transplant, each strain was inoculated into 50 mL of Marine Broth (Condalab, Spain) and incubated at 22°C for up to 48 h to reach a concentration of $10^8$ CFU/mL. After incubation, the entire culture was centrifuged at 4,000 rpm for 10 min, the supernatant discarded, and the resulting pellet resuspended in 16.7 mL of ASW. To produce gnotobiotic clams, the three bacterial suspensions were then combined to give a final volume of 50 mL ASW containing $10^8$ CFU/mL *Vibrio diazotrophicus*, $10^8$ CFU/mL *Shewanella colwelliana,* and $10^8$ CFU/mL *Halomonas alkaliphila*. This mixture was then added directly to the two treatment tanks. The remaining two tanks were used as controls, and 50 mL of ASW was added. One hour after bacterial inoculation, three clams from each tank (transplanted and controls) were collected, sacrificed, and pooled (one sample per tank) for the assessment of microbial load and 16S rRNA analysis (T4, Table 1). The contents of each tank were transferred to a new aquarium containing 2.5 L of ASW (a total of 3 L of ASW per aquarium with nine transplanted or control clams) and kept at 25°C. After 6 h (T5, Table 1) and 22 h (T6, Table 1), three clams from each tank were collected, sacrificed, and pooled (one sample per tank) for microbial load assessment and 16S rRNA analysis.

### Clam collection and dissection

The clams were collected at the time points listed in Table 1. For T0 and T1, three samples of three clams each were collected from the same aquarium and processed separately. For T2 and T3, three clams were collected from each aquarium (3 treatment and 1 control – 4 samples in total), sacrificed, and pooled as follows. For each clam, the adductor muscle was cut, the valves were opened, and the entire wet body was collected in a stomach bag together with the extrapallial fluid using sterile tweezers and a pipette. Three clams were collected in the same stomacher bag and homogenized together by

adding 9 volumes of phosphate buffered saline (PBS) (Sigma-Aldrich, Germany) of the sample weight for 2 min at maximum speed using the Stomacher 3500 (VWR, Italy). To assess the microbial load, samples were serially diluted in PBS, plated on MA (Condalab, Spain), and incubated at 22°C for 48 h.

For subsequent RNA extraction, 2 mL of the homogenized samples were centrifuged at 4000 rpm for 2 min, and the pellet was stored at −80°C.

## RNA extraction, amplification, and sequencing

For RNA extraction, 0.5 µm glass beads were added to the previously stored pellet, which was then processed using the RNeasy Mini Kit (Qiagen, Germany) according to the manufacturer's instructions. RNA concentration was measured using a NanoDrop ND-1000 spectrophotometer (Thermo Scientific, USA). For microbiota characterization, 1 µg of RNA was reverse transcribed into cDNA using the SuperScript IV First-Strand Synthesis System (Invitrogen, USA).

The 16S rRNA gene was amplified using two universal 16S primers (forward primer, UniF 5′-GTGSTGCAYG GYTGTCGTCA-3′ and reverse primer, UniR 5′-ACGTCRTCCM-CACCTTCCTC-3′) (38). The 16S rRNA gene of *Endozoicomonas* spp. was amplified using an *Endozoicomonas*-specific primer set including a reverse primer (En771R: 5′-TCAGTGT-CARRCCTGAGTGT-3′) and a bacterial universal forward primer (27F: 5′-AGAGTTGATC MTGGCTCAG-3′) (39). End-point PCRs were performed in a total of 20 µl on Mastercycler nexus SX1 (Eppendorf, Hamburg) using DreamTaq PCR Master Mix (Thermo Scientific, USA). Reaction mixtures consisted of 0.5 µL of each primer, 10 µL PCR Master Mix, 7 µL water, and 2 µL DNA (or cDNA) template. PCR conditions included 1 cycle of initial denaturation at 94°C for 2 min, followed by 35 cycles of denaturation at 94°C for 20 s, annealing at 56°C for 30 s, extension at 72°C for 30 s, and a final extension at 72°C for 7 min. PCR products were observed by 1.5% agarose gel electrophoresis.

## 16S rRNA sequencing and statistical analysis

For 16S rRNA gene amplicon sequencing, library preparation and sequencing of the V3–V4 hypervariable regions of the bacterial 16S rRNA gene were performed at BMK Gene (Germany). Sequencing was performed on an Illumina Novaseq 6000. Raw data were processed in order to remove adapters, select reads lengths, and remove low-quality reads using Fastp (40), Trimmomatic v0.33 (41), and cutadapt 2.7.8 (42). Cleaned reads were then processed for downstream analyses using R software with dada2 (43) to identify Amplicon Sequence Variants (ASV) and obtain a taxonomic identification using silva_nr99_v138.1_wSpecies_train_set.fa as database. Bacterial genera with a relative abundance greater than 1% were included in the analyses. The association between time points and genus-level composition was assessed using MaAsLin2, which utilizes general linear models (44). Biodiversity metrics were estimated using *phyloseq* (45) and *vegan* packages in R, with Bray-Curtis distance used for Beta-Diversity. Graphical representations, including PCoA and bar plots, were generated using the *ggplot2* and *RColorBrewer* packages (46, 47). ANOVA tests were conducted using the compare_means function from the *ggpubr* package to assess the variance in the abundance of each species over time.

## RESULTS

### Generation of germ-free clams

In this study, we developed a protocol for the microbiological sterilization of adult Manila clams by administering a mixture of antibiotics in the water (Fig. 1). After 5 days of acclimation at 25°C (T1), the clams were divided into four tanks: three of which were treated with a mixture of five antibiotics, while the remaining tank was used as a control. The four tanks were kept at 25°C for 24 h.

To assess the efficacy of the antibiotic treatment, we first monitored microbial growth from T0 (when the clams were purchased from the hatchery) to T3 (after two doses of

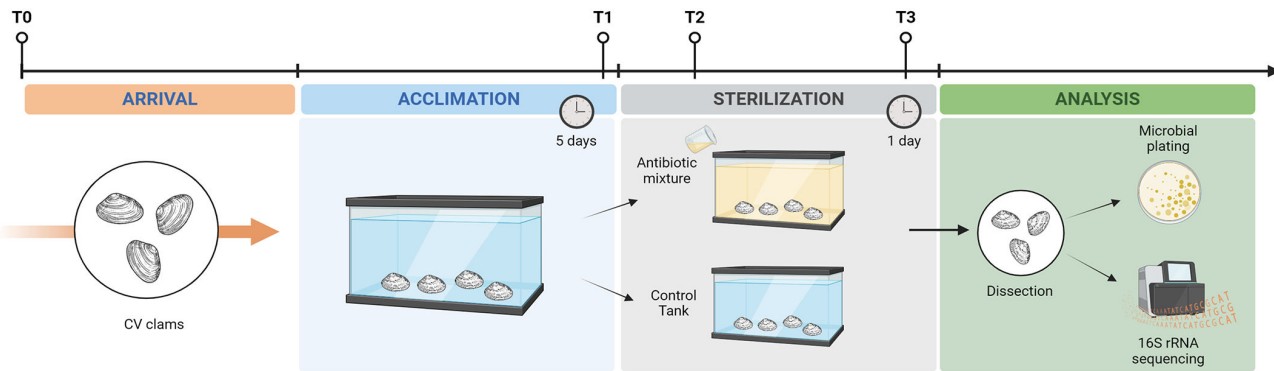

**FIG 1** Schematic diagram of the experimental protocol for the production of germ-free clams (CV: conventional clams; T0: arrival; T1: 5 days post-acclimation; T2: 6 h post-antibiotic administration; T3: 20 h post-antibiotic administration). Image created with biorender.com.

antibiotics) by plating the clam homogenate on Marine Agar (MA) culture medium, a non-selective medium commonly used to grow and isolate a wide variety of heterotrophic marine bacteria. Starting from a total microbial load of $10^6$ CFU/mL (T0 – $n = 3$ clams), microbial growth decreased over time and no bacterial growth was detected after two doses of antibiotics. One dose of antibiotics resulted in a two-log reduction in microbial load (T1, Fig. 2A).

To assess the presence of viable but nonculturable bacteria in the treated clams, we also performed 16S rRNA sequencing of RNA extracted from the clam homogenate. A total of seven samples were sequenced on Illumina Novaseq6000 sequencing platform, generating 8,527,652 pair raw reads. These PE reads were processed for quality control, assembly, and data filtration, which yielded 6,402,125 clean reads. A minimum of 50,186 clean reads were generated for each sample, and the average data output per sample was 72,751 clean reads. Taxonomic analyses at both the phylum and genus level revealed a significant decrease in the microbial community of clams following antibiotic treatment, while control clams (i.e., not treated with antibiotics) showed an increase in the number and type of bacterial genera (Fig. 2B, Fig. S2, Table S1). After acclimation (T1), the clam microbiome was characterized by a total of 26 bacterial genera, with the predominant presence of *Endozoicomonas* (82.9%), followed by *Pseudomonas* spp. (4.1%), *Bacillus* spp. (2.5%), *Umboniibacter* spp. (2%), *Salinirepens* spp. (1.7%), *Oleiphilus* spp. (1.3%), *Stenotrophomonas* spp. (1.2%), and 19 other bacterial genera, representing less than 1% each (Table S1, Fig. 2B). The administration of the antibiotic mixture caused a significant decrease in the diversity of the clam microbial community (Fig. S2), which was dominated by *Endozoicomonas* spp. (98%) (Fig. 2B). The abundance of *Bacillus* spp. decreased to 1%, and traces of five other bacterial genera were also detected, but their abundance was less than 0.01% (Table S1, Fig. 2B). However, the prevalence of 24 bacterial genera was observed in the control aquarium. In particular, the dominance of the genus Endozoicomonas, which was prominent at the end of acclimation and resistant to antibiotic treatment, decreased to 46% in untreated clams. This reduction was accompanied by an enrichment of other bacterial genera, including *Malaciobacter* spp. (8%), *Umboniibacter* spp. (7.4%), *Neptuniibacter* spp. (6%), *Alteromonas* spp. (6%), *Pontibacterium* spp. (5%), *Salinirepens* spp. (4.5%), *Bacillus* spp. (2%), Marinomonas spp. (2%), *Aliiroseovarius* spp. (1.7%), *Marinobacter* spp. (1.4%), *Lentibacter* spp. (1.3%), *Acinetobacter* spp. (1%), *Sulfitobacter* spp. (1%), and 10 other bacterial genera, each representing less than 1% (Table S1, Fig. 2B).

Overall, our results demonstrate that the protocol developed to obtain germ-free clams resulted in the near-complete depletion of the clam microbiome, with almost exclusively *Endozoicomonas* spp. persisting in the antibiotic-treated clams. To classify the

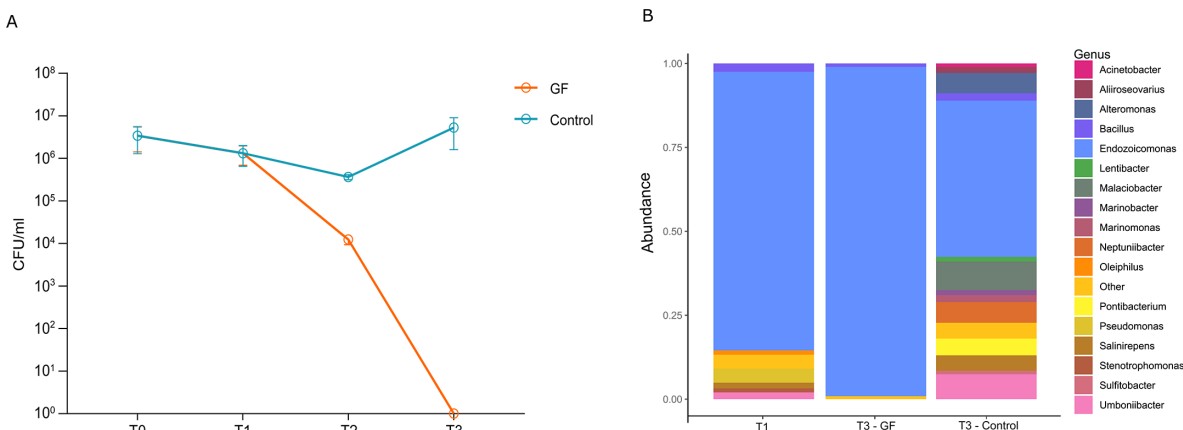

**FIG 2** (**A**) Microbial load (CFU/mL) detected in clams after antibiotic treatment (GF) and in control animals (T0: arrival; T1: 5 days post-acclimation; T2: 6 h post antibiotic administration; T3: 20 h post antibiotic administration). (B) Taxonomic composition of relative microbiome abundance at genus level of acclimated clams (T1), antibiotic treated (T3—GF) and control clams (no antibiotic treatment; T3—Control). Bar plot of significant genera (genera with relative abundance <0.1% are grouped as "other").

antibiotic-resistant *Endozoicomonas* at the species level, we performed end-point PCR and Sanger sequencing of the 16S rRNA gene from treated clams using *Endozoicomonas*-specific primers. The resulting sequence was identified as *Endozoicomonas elysicola* (Fig. S3, File S2).

### Generation of gnotobiotic clams

#### *Selection of bacterial species for the mock community*

In order to carry out the transplant experiment, we sought to create a mock community consisting of known concentrations of defined bacterial species, capable of colonizing bivalve molluscs. To achieve this, six acclimated clams (T1) were collectively crushed and the resulting homogenate was plated on selective and differential media. Following incubation, three colonies were randomly selected based on their unique morphology on MA for 16S rRNA gene sequencing for species-level classification. The species were finally identified as *Vibrio diazotrophicus*, which is characterized by small round, smooth, opaque white colonies with a halo, *Shewanella colwelliana*, showing round, smooth, pink colonies, and *Halomonas alkaliphila*, which has round, smooth, convex, and creamy white colonies.

#### *Microbiome transplant*

To proceed with the transplant of the mock community, we first treated the Manila clams with the antibiotic regimen as described above (Fig. 1). Twenty hours after antibiotic administration (T3), clams were depurated from antibiotics for 2 h before being transferred to small tanks with fresh ASW (Fig. 3). The microbial mock suspension, consisting of the three selected bacterial species (final volume: 50 mL; [C] = $10^8$ CFU/mL per species), was added directly to the water containing the clams. The control tanks received an equal volume of ASW (Fig. 3). After 1 h of incubation (T4), the entire contents of the small tanks (including clams and water, with or without bacterial suspension) were transferred to new tanks containing fresh ASW (total volume = 3 L per tank).

To assess the efficacy of the microbiome transplant, we monitored bacterial growth throughout the experiment (Fig. 4A). No bacterial growth was detected after antibiotic treatment (T4, Fig. 4A). In the transplanted clams, a rapid and significant increase in microbial load was observed as early as 1 h after transplant (T4 – $10^4$ CFU/mL), with a final concentration of $10^6$–$10^5$ CFU/mL (T5 and T6, respectively). In contrast, control clams (i.e., animals that received antibiotic treatment but no microbiome transplant) showed

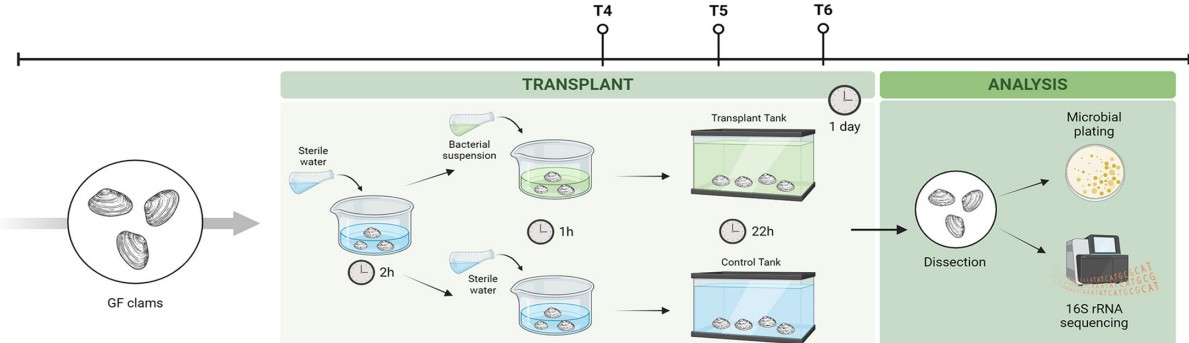

**FIG 3** Schematic diagram of the experimental protocol for the microbiota transplant in germ-free (GF) clams (T4: 1 h post-transplant; T5: 6 h post transplant; T6: 22 h post-transplant). Image created with biorender.com.

a detectable microbial load at 8 h after transplant (T5 – $10^2$ CFU/mL), reaching a final concentration of $10^6$ CFU/mL at T6 (Fig. 4A).

To determine the effect of microbial transplant on the composition of the bacterial community, we performed 16S rRNA amplicon sequencing of the transplanted and control individuals. A total of 22 samples were sequenced on Illumina Novaseq6000 sequencing platform, generating 10,943,861 pair reads. A minimum of 220,468 clean reads were generated for each sample, and the average data output per sample was 497,448 clean reads. Principal Coordinates Analysis (PCoA) revealed a clear clustering between transplanted and control individuals (Principal Component 2—PC2; Fig. 4B). This suggests that a differentiation of the clam microbiome has occurred following microbial transplant. While the control samples (i.e., non-transplanted clams) maintain a stable association with *Endozoicomonas* spp. (Fig. 4C; Fig. S4), the bacteria that contribute to the post-transplant differentiation are, indeed, the three inoculated bacterial species that were detected and significantly increased only in the transplanted clams, both by 16S rRNA amplicon sequencing and by plating (Fig. 4D through E; Table S3). In particular, *H. alkaliphila* and *S. colwelliana* colonized the transplanted individuals more efficiently, while *V. diazotrophicus* was recovered at lower levels (Fig. 4D through E, Table 2). Interestingly, all transplanted species showed a significant increase in recipient clams, which was accompanied by a significant decrease in six bacterial species, suggesting outcompetition by the transplanted species (Fig. 4F; Tables S3 and S4).

## DISCUSSION

The generation of gnotobiotic animals represents a powerful strategy to study the mechanisms underlying the relationship between animal hosts and their microbiome at different levels, allowing the dissection of physiological response down to molecular processes. Since the late 19th century, various animals have been sterilized for research purposes. During this time, basic research on host-microbe interactions has been conducted with germ-free and gnotobiotic model organisms, from invertebrates (*C. elegans*, *Drosophila melanogaster*, etc.) and vertebrates such as zebrafish and mice.

However, the nature of host-microbe relationships varies greatly across hosts and ecosystems, as many of them are highly dependent on the environment. Therefore, it is essential to study dynamics and processes in non-model organisms and complex systems. Here, we present a new method to generate gnotobiotic bivalves using the Manila clam *Ruditapes philippinarum* as a model species. This species, which is found in lagoons and river deltas, is of great importance from an environmental standpoint, as it provides a range of ecosystem services (29). Additionally, it is a valuable economic resource for global aquaculture, offering benefits (e.g., economic and societal) to local communities of producers, which are often centered around bivalve farms.

In this work, we have developed a protocol for microbiome depletion and transplant on adult clams. The microbiome of the treated clams consisted mainly of

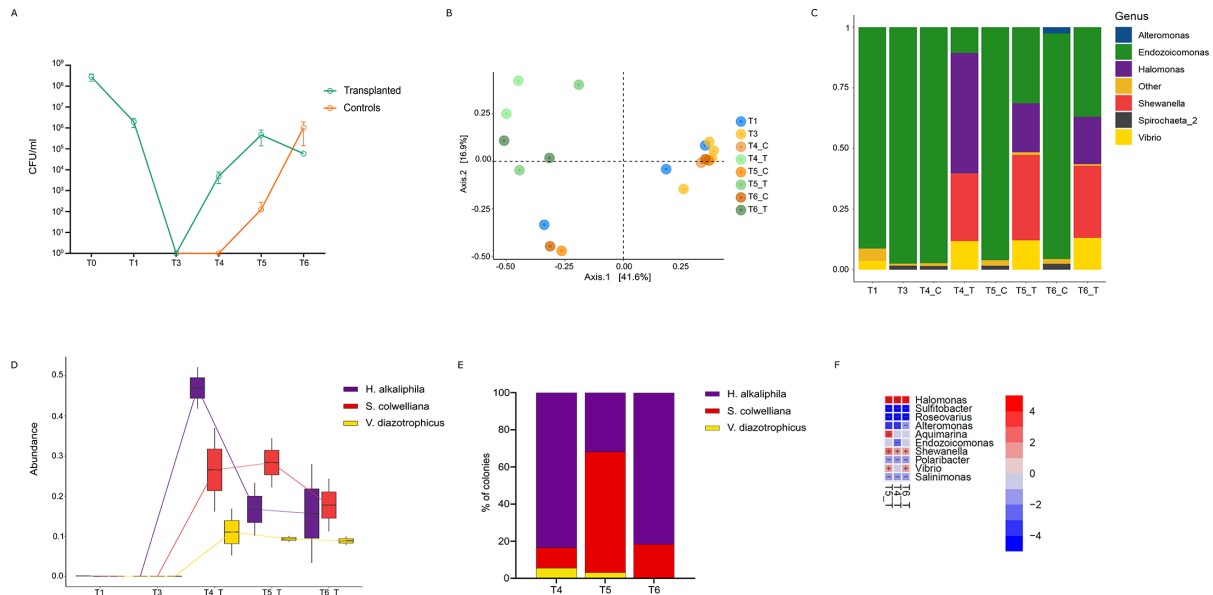

**FIG 4** (**A**) Microbial load (CFU/mL) detected in clams throughout the experiment. The description of the time points is given in Table 1. (**B**) PCoA (at ASV level) illustrating sample clustering based on bacterial treatment and time (C: control clams; T: transplanted clams). (**C**) Taxonomic composition of relative microbiome abundance at genus level. Genera with relative abundance <1% are grouped as "Other." (**D, E**) Relative abundance of the three bacterial species selected for microbial transplant obtained as determined by 16S rRNA sequencing (**D**) and microbial plating on MA (**E**). (**F**) Maaslin2 results on transplanted clams using T3 as fixed effect.

*Endozoicomonas* spp., followed by *Pseudomonas*, *Umbonibacter,* and a few other bacterial genera (Fig. 2B). The microbiome-depletion protocol successfully reduced all bacterial genera, with the exception of *Endozoicomonas* spp., which was only detected by 16S rRNA amplicon sequencing and consequently identified as *Endozoicomonas elysicola* by Sanger sequencing. No bacterial growth was observed after 20 h of antibiotic treatment, indicating that *E. elysicola* was not culturable on MA. This is noteworthy as *Endozoicomonas* spp. have been shown to be cultivable on general media such as MA (48, 49), suggesting that the strain present in the treated clams may have specific nutritional requirements and/or be difficult to isolate. *Endozoicomonas* spp. are prevalent symbionts in a variety of marine hosts, including corals (50), and other cnidarians (51), sponges (52), gorgonians (53), worms (54), fish (55), tunicates (56), and molluscs (57). They generally reside in aggregates within the host endodermal tissues (50). For these reasons, it has not always been easy to isolate *Endozoicomonas* from host tissues (58), and despite its associations with numerous hosts in oceans worldwide, the functional role of *Endozoicomonas* remains unclear. Further studies are needed to investigate where *Endozoicomonas* resides (i.e., in organs or tissues) using imaging techniques, which have been shown to be effective in revealing the spatial distribution of such species (49, 50). This would be important information for the development of targeted treatments to effectively deplete the species.

**TABLE 2** Relative abundance of the three bacterial species selected for microbial transplant obtained as determined by 16S rRNA sequencing and microbial plating of transplanted clams

| Transplanted species | Relative abundance | | | | | |
|---|---|---|---|---|---|---|
| | 16S amplicon sequencing | | | Microbiological plating | | |
| | T4 | T5 | T6 | T4 | T5 | T6 |
| *Halomonas alkaliphila* | 49.6 | 20.1 | 19.5 | 83.4 | 31.8 | 81.6 |
| *Shewanella colwelliana* | 27.8 | 35.2 | 19.529.6 | 11.1 | 65.1 | 18.4 |
| *Vibrio diazotrophicus* | 11.8 | 12.1 | 13.1 | 5.5 | 3.1 | 0 |

The protocol we developed for microbiome transplantation on adult Manila clams is based on the inoculation of a mock community consisting of three bacterial species. The method proved to be effective in transplanting the three bacterial species although the final abundances in the recipient clams differed. Indeed, the recipient animals showed a distinct microbiome profile in comparison to the control animals (Fig. 4B), with a significant increase in the abundance only of the transplanted species (Fig. 4C and D). This result was also confirmed by classical microbiological plating (Fig. 4E).

Transplantation appears to be effective as early as 1 h after inoculation of the bacterial communities, and the species were still found after 20 h. Notably, a single microbiome transplantation appears to lead to a slight decrease of the transplanted species over time, with the exception of *V. diazotrophicus*, which appears to efficiently colonize the clams and be stable over time (Fig. 4C, D and F). It would be interesting to test the effect of multiple transplantations on recipient clams to further extend the efficacy of transplantation, as well as to test the efficiency of the protocol in the longer term. It is also noteworthy that starting with a microbiome-depleted organism, rather than a completely sterile one, still allowed efficient transplantation of the desired microbial community. This finding is consistent with results from other host-microbe systems biology studies using microbiome-depleted organisms (59, 60) and is particularly relevant when working with non-model organisms or in experimental settings where achieving complete sterility is challenging. It also highlights the practicality of this approach for field applications where strict microbial sterility is not always feasible. In addition, the transplantation method resulted in a significant reduction in the relative abundance of *E. elysicola*, the dominant bacterial species resistant to antibiotic treatment, with all three transplanted species contributing to its outcompetition (Fig. 4C). These results are particularly intriguing in the context of the need to reduce antibiotic use, including in aquaculture systems. They demonstrate that harnessing the higher fitness of symbionts, especially beneficial symbionts, could be a more effective strategy for outcompeting pathogens. This paves the way for targeted treatments based on the administration of probiotics and beneficial symbionts to animals in aquaculture facilities.

Of note, the baseline temperature of 25°C used for both acclimation and transitions between experimental phases was chosen to reflect the average summer temperature of the Venice Lagoon—one of the world's most important farming regions for *R. philippinarum* and a routine bivalve sampling site. By validating our protocol at this temperature, we aimed to ensure its applicability to real field conditions beyond the laboratory. Furthermore, given the increasing use of bivalve molluscs in studies of climate change and thermal stress (32, 61), 25°C serves as an ideal baseline that can be raised to investigate thermal stress responses. Confirming the efficacy of microbiome transplantation at this temperature was, therefore, essential for the wider application of the protocol.

Although our technique is highly effective in significantly reducing the microbiome of clams and generating gnotobiotic animals, technical challenges need to be addressed to further advance the study of gnotobiotic bivalves. First, improved methods to efficiently isolate the aquaria (e.g., seals, sterilized pumps, etc.) as well as automated water filtration methods need to be developed to completely eliminate environmental contamination. Second, testing our protocols on clams of different origins, possibly carrying different microbiomes, would allow us to test the applicability of our techniques under different ecological conditions. Finally, the definition of a sterile diet could represent an added value allowing different laboratories conducting microbiome research on bivalves to standardize the nutritional regimes.

In conclusion, this newly developed method for generating gnotobiotic bivalves will significantly enhance the analysis of microbial impacts on animal health in marine ecosystems, thereby expanding the potential of gnotobiotic research—particularly in studies aimed at identifying causal relationships. With rapid environmental changes increasingly affecting marine ecosystems, this technique offers a critical opportunity to

deepen our understanding of how microbes influence stress resilience in molluscs and explore their potential to enhance adaptive capacity.

## ACKNOWLEDGMENTS

This work was supported by the project PRIMECLAMS from "Fondazione Cassa di Risparmio di Padova e Rovigo" (CUP: C13C21000160005) awarded to L.B.

## AUTHOR AFFILIATION

[1]Department of Comparative Biomedicine and Food Science, University of Padova, Padua, Italy

## AUTHOR ORCIDs

Luca Bargelloni http://orcid.org/0000-0003-4351-2826
Maria Elena Martino http://orcid.org/0000-0001-5038-5605

## FUNDING

| Funder | Grant(s) | Author(s) |
| --- | --- | --- |
| Fondazione Cassa di Risparmio di Padova e Rovigo (Fondazione Cariparo) | C13C21000160005 | Luca Bargelloni |

## AUTHOR CONTRIBUTIONS

Marialaura Gallo, Data curation, Investigation, Methodology, Writing – original draft, Writing – review and editing | Andrea Quagliariello, Data curation, Formal analysis, Writing – review and editing | Giulia Dalla Rovere, Methodology, Writing – review and editing | Federica Maietti, Methodology | Barbara Cardazzo, Writing – review and editing | Luca Peruzza, Methodology, Writing – review and editing | Luca Bargelloni, Funding acquisition, Project administration, Resources, Writing – review and editing, Conceptualization | Maria Elena Martino, Conceptualization, Data curation, Investigation, Methodology, Supervision, Writing – original draft, Writing – review and editing

## DATA AVAILABILITY

Sequencing data are available in the NCBI BioProject database (BioProject ID PRJNA1223360).

## ADDITIONAL FILES

The following material is available online.

### Supplemental Material

**Fig. S1 (Spectrum01189-24-S0001.png).** Taxonomic composition of relative microbiome abundance at phylum level of acclimated clams (T1), antibiotic treated (T3-GF), and control clams (no antibiotic treatment; T3-Control).
**Fig. S2 (Spectrum01189-24-S0002.tif).** Observed and Shannon indices were estimated for the two time-points.
**Fig S3 (Spectrum01189-24-S0003.png).** 1.5% agarose gel showing end-point PCR amplification products from experimental samples and controls.
**Fig. S4 (Spectrum01189-24-S0004.png).** Taxonomic composition of relative microbiome abundance of experimental samples at phylum level.
**Supplemental material (Spectrum01189-24-S0005.docx).** Supplemental figure legends.
**Table S1 (Spectrum01189-24-S0006.docx).** Relative abundances of all bacterial genera detected in the experimental samples through 16S rRNA sequencing.

**Table S2 (Spectrum01189-24-S0007.docx).** Sanger sequence of 16S rRNA gene of *Endozoicomonas spp*. detected in antibiotic-treated clams.

**Table S3 (Spectrum01189-24-S0008.docx).** Statistical results showing p-values and adjusted p-values from ANOVA tests assessing variance in species abundance over time.

**Table S4 (Spectrum01189-24-S0009.docx).** Relative abundances of all bacterial genera detected in the experimental samples through 16S rRNA sequencing.

## Open Peer Review

**PEER REVIEW HISTORY (review-history.pdf).** An accounting of the reviewer comments and feedback.

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
