## [Reviewer comments · Microbiology Spectrum]

Microbiology Spectrum

Generating gnotobiotic bivalves: a new method on Manila clam (*Ruditapes philippinarum*)

Marialaura Gallo, Andrea Quagliariello, Giulia Dalla Rovere, Federica Maietti, Barbara Cardazzo, Luca Peruzza, Luca Bargelloni, and Maria Elena Martino

Corresponding Author(s): Maria Elena Martino, Universita degli Studi di Padova

Review Timeline:

Submission Date:	May 13, 2024
Editorial Decision:	February 1, 2025
Revision Received:	February 18, 2025
Editorial Decision:	April 30, 2025
Revision Received:	May 8, 2025
Accepted:	June 30, 2025

Editor: Francisco Uzal

Reviewer(s): Disclosure of reviewer identity is with reference to reviewer comments included in decision letter(s). The following individuals involved in review of your submission have agreed to reveal their identity: Zhen Zhang (Reviewer #1)

Transaction Report:

DOI: <https://doi.org/10.1128/spectrum.01189-24>

Re: Spectrum01189-24 (Generating gnotobiotic bivalves: a new method on Manila clam (*Ruditapes philippinarum*))

Dear Dr. Maria Elena Martino:

Thank you for the privilege of reviewing your work. Below you will find my comments, instructions from the Spectrum editorial office, and the reviewer comments.

Revision Guidelines

Sincerely,
Francisco Uzal
Editor
Microbiology Spectrum

Reviewer #1 (Comments for the Author):

The work is good but needs major modifications (see attached).

Reviewer #2 (Comments for the Author):

In the paper "Generating gnotobiotic bivalves: a new method on Manila clam (*Ruditapes philippinarum*)", the authors present a method to generate gnotobiotic marine models to enable future investigations of marine invertebrate microbes.

The manuscript fits the scope of the journal and is interesting.

The authors' clear acknowledgment of their study's limitations in the discussion section is commendable. Overall the manuscript is well written.

L30: I suggest modifying this statement. After reading, *Ruditapes philippinarum* is stated as a sentinel organism for environmental pollution but not climate change.

L47: I'm not fully convinced that "the ease of data collection" is the primary driver for the "overemphasis on the role of microbiomes in animal health". A more plausible explanation might lie in the complexities associated with assessing the functionalities of the microbiome. I would recommend revising this segment.

L61-70: 10 lines with no references. I strongly recommend the authors to add references.

The introduction mentions that gnotobiotic organisms have been created before and that these models are useful for investigating the influence of microorganisms on animal health. However, it lacks specific examples. Could the authors incorporate in the introduction examples of examples from prior studies that used gnotobiotic models to research the microbiome? This would enhance the readers' comprehension of the concept.

L104-116: "The final acclimation temperature (25{degree sign}C) reflected the temperature of the subsequent experimental phases." it is unclear why the temperature needs to be 25C for the experimental phases.

L113: Is there any bacteria present in the food used in the experiment?

L142: how were the antibiotics validated?

L155: remove the extra space between morphological and differences. It also appears there might be additional instances of double spacing throughout the manuscript, please check.

L178-180: The reasoning behind the composition of the bacterial suspension, which includes these three specific bacteria, isn't clearly explained. I suspect it's based on the findings. Even though this is the methodology section, I would recommend the authors to incorporate a statement clarifying the origin of this selection.

L225-231: the 16S rRNA sequencing analysis paragraph lacks of details. The authors should include all the details linked to data analysis as well as provide the scripts.

Not a single statistical test was realized. I strongly recommend the authors to conduct statistical analysis on their results.

The authors should provide the repository information where the sequencing data has been stored and can be accessed. This is crucial for the transparency and reproducibility of the research.

The transplantation experiment's objective is not clearly defined in the manuscript. I suggest that the authors elaborate on this in the methods section, and also provide context in both the introduction and discussion sections. This will help readers understand its relevance and importance to the overall study.

The authors affirm that they successfully developed a gnotobiotic clam despite the detection of bacteria. I propose that in the discussion, the authors refer to previous studies with analogous results to reinforce their findings. My intention is not to imply that the clams are not gnotobiotic, but rather to preempt any potential skepticism on the part of the reader.

L372-377: very interesting.

Dear authors

The research work entitled “Generating gnotobiotic bivalves: a new method on Manila clam (*Ruditapes philippinarum*)”, is crucial and an interesting research work which highlighted the development of a model animal for the study of the microbiota diversity and functions of bivalve mollusks. I appreciate for your effort.

General comments

The manuscript needs a major revision to make the paper a significant for the scientific community. Therefore, I provided my suggestion and comments below:

- ✓ The abstract part was written in a very shallow. So it is better to rewrite this part by including the main methodological aspects and the findings.
- ✓ Introduction parts should mainly focused on the development of aquatic model animals in the study of climate change and its effect on aquatic microbiome communities and aquatic animals, as change in the microbiome communities can be used as a biomarkers to assess the condition of the aquatic animals and to take measures.
- ✓ In the materials and methods it should be written in a scientific manner and in detail. During formation of germ free clam the environment should be free of bacterial contamination. And bacteria contamination assessment should be restricted. In addition during bacterial transplantation, authors select specific bacteria but the nature and properties of these bacteria were not explained in detailed.
- ✓ Parameters of water quality should be added.
- ✓ The procedure on transplant of the mock community should be written in detailed
- ✓ The volume of the combined antibiotics given at a time should be mentioned.
- ✓ The selection of the mock microbiome community needs more clarifications.

Specific concerns and comments:

- It is better to add the water parameters including dissolved oxygen and nitrite levels during clam maintenance and acclimation.
- Antibiotic treatment: what was the volume of the mixture of antibiotics provided during the trails?

- Better to add the efficiency of the antibiotics
- The antibiotics are provided twice in 6 hours differences, why authors choose 6 hour interval?
- During clam crushing, it is better to add the value of the maximum speed of Stomacher® 3500?
- After crushing of clam, do authors centrifuge the homogenate? And authors take 100 ul of homogenate is it the supernatant part or the whole?
- During microbiome transplantation do clams fed with commercial diet?
- Source of the bacteria used for to produce *Vibrio diazotrophicus*, *Shewanella colwelliana* and *Halomonas alkaliphila* and the colonization capability of these bacteria for bivalves should be explained in detailed.
- Line 183-185 Do authors use the culture water or the clam contents to transplant the bacteria? Need detailed explanation on this?
- Line 221-223, should be transferred to the subtitle “16S rRNA sequencing analysis”
- 16S rRNA sequencing analysis should be written in depth.
- Why the abundance of bacteria sampled from T1 and control T3 are huge difference?
- Line 284-285, authors select three colonies of bacteria based on their unique morphology for 16S rRNA analysis, is morphological characteristics is important to determine the function of the bacteria?
- Add the 16S rRNA results of the bacteria composition at the phylum level.

Reviewer #1 (Comments for the Author):

The work is good but needs major modifications (see attached).

Dear authors,

The research work entitled “Generating gnotobiotic bivalves: a new method on Manila clam (*Ruditapes philippinarum*)”, is crucial and an interesting research work which highlighted the development of a model animal for the study of the microbiota diversity and functions of bivalve mollusks. I appreciate for your effort.

General comments

The manuscript needs a major revision to make the paper a significant for the scientific community. Therefore, I provided my suggestion and comments below:

- The abstract part was written in a very shallow. So it is better to rewrite this part by including the main methodological aspects and the findings.

R: We thank the reviewer for their comments. We agree that increasing the level of detail in the abstract has improved the overall description. Accordingly, we have significantly reworded and revised the Abstract section to follow the reviewer’s suggestion (lines 8-24).

- Introduction parts should mainly focused on the development of aquatic model animals in the study of climate change and its effect on aquatic microbiome communities and aquatic animals, as change in the microbiome communities can be used as a biomarkers to assess the condition of the aquatic animals and to take measures.

R: We have revised both the Introduction and Discussion sections to address the reviewers’ suggestions (Lines 69-75; 431-436).

- In the materials and methods it should be written in a scientific manner and in detail. During formation of germ free clam the environment should be free of bacterial contamination. And bacteria contamination assessment should be restricted. In addition during bacterial transplantation, authors select specific bacteria but the nature and properties of these bacteria were not explained in detailed.

- Parameters of water quality should be added.

- The procedure on transplant of the mock community should be written in detailed

- The volume of the combined antibiotics given at a time should be mentioned.

- The selection of the mock microbiome community needs more clarifications.

R: We have provided detailed responses to the five points above in the Specific Concerns section below, as they were also listed in this section. Please see the section below for our responses. We fully agree with the reviewer regarding the importance of preventing bacterial contamination in the production of germ-free clams. The Materials and Methods section includes a dedicated paragraph detailing the procedures and considerations used prior and during the antibiotic treatment (Lines 131-136).

Specific concerns and comments:

- It is better to add the water parameters including dissolved oxygen and nitrite levels during clam maintenance and acclimation.

R: We have added the requested details in Lines 113-128. We did not measure the nitrite levels but water changes were made every 48 hours to prevent the build-up of toxic compounds.

- Antibiotic treatment: what was the volume of the mixture of antibiotics provided during the trails?

R: We have added the requested details in Lines 160-165.

- Better to add the efficiency of the antibiotics

R: We have added the requested details in Lines 144-157.

- The antibiotics are provided twice in 6 hours differences, why authors choose 6 hour interval?

R: We tested different time intervals for the second antibiotic inoculation and, based on preliminary experiments, a six-hour interval proved to be the most effective in achieving maximum bacterial load reduction (Lines 144-146).

- During clam crushing, it is better to add the value of the maximum speed of Stomacher® 3500?
R: We have added the requested details in Line 177.

- After crushing of clam, do authors centrifuge the homogenate? And authors take 100 ul of homogenate is it the supernatant part or the whole?

R: The homogenate is not centrifuged after homogenisation, 100 µl are collected directly from the filtrate obtained in the Stomacher bag.

- During microbiome transplantation do clams fed with commercial diet?

R: The clams were fed once at the beginning of the transplantation phase with the commercial diet described in the Methods section. We have added further details of the feeding schedule in lines 201-202.

- Source of the bacteria used for to produce *Vibrio diazotrophicus*, *Shewanella colwelliana* and *Halomonas alkaliphila* and the colonization capability of these bacteria for bivalves should be explained in detailed.

R: We have expanded the 'Selection of the mock community' section to include the rationale for the selection of the candidate strains, together with additional methodological details of their isolation (Lines 173-184).

- Line 183-185 Do authors use the culture water or the clam contents to transplant the bacteria? Need detailed explanation on this?

R: We have expanded the 'Transplant of the mock community' section to include additional methodological details of bacterial culture and inoculation (Lines 202-216).

- Line 221-223, should be transferred to the subtitle "16S rRNA sequencing analysis"

R: We have transferred the paragraph to the suggested section.

- 16S rRNA sequencing analysis should be written in depth.

R: We have expanded the Methods section related to 16S rRNA sequencing (Lines 251-264).

- Why the abundance of bacteria sampled from T1 and control T3 are huge difference?

*R: If the reviewer is referring to Figures 2A,B and 4A,C, we observed a significant decrease in bacterial abundance following antibiotic treatment (T3) compared to the post-acclimation phase (T1) (see Table 1), where bacterial abundance was 10^6 CFU/ml (see Fig. 2 and 4). This suggests that the antibiotic treatment was effective in eliminating most bacteria, with the exception of *Endozoicomonas elysicola*. We added results of statistical analyses in the Methods section and Fig. S2.*

- Line 284-285, authors select three colonies of bacteria based on their unique morphology for 16S rRNA analysis, is morphological characteristics is important to determine the function of the bacteria?

R: As mentioned above, we have expanded the 'Transplantation of the mock community' section to include additional methodological details on the selection of bacterial strains for transplantation. Specifically, morphological characteristics were used as a rapid discriminatory variable during microbiological plating following microbiome transplantation, allowing verification of the simultaneous presence of all three bacterial strains (Lines 202-216).

- Add the 16S rRNA results of the bacteria composition at the phylum level

R: We have added the requested analysis in Supplementary Figures S2 and S3 (Lines 288, 346).

Reviewer #2 (Comments for the Author):

In the paper "Generating gnotobiotic bivalves: a new method on Manila clam (*Ruditapes philippinarum*)", the authors present a method to generate gnotobiotic marine models to enable future investigations of marine invertebrate microbes.

The manuscript fit the scope of the journal and is interesting.

The authors' clear acknowledgment of their study's limitations in the discussion section is commendable.

Overall the manuscript is well written.

L30: I suggest modifying this statement. After reading, *Ruditapes philippinarum* is stated as a sentinel organism for environmental pollution but not climate change.

R: We agree with the reviewer and modified the sentence accordingly (Line 34).

L47: I'm not fully convinced that "the ease of data collection" is the primary driver for the "overemphasis on the role of microbiomes in animal health". A more plausible explanation might lie in the complexities associated with assessing the functionalities of the microbiome. I would recommend revising this segment.

R: We agree with the reviewer and modified the sentence accordingly (Lines 50-54).

L61-70: 10 lines with no references. I strongly recommend the authors to add references.

R: We have added relevant references to support the points mentioned throughout the text (Lines 66-69).

The introduction mentions that gnotobiotic organisms have been created before and that these models are useful for investigating the influence of microorganisms on animal health. However, it lacks specific examples. Could the authors incorporate in the introduction examples of examples from prior studies that used gnotobiotic models to research the microbiome? This would enhance the readers' comprehension of the concept.

R: We have expanded the overview of gnotobiology in host-microbe interactions and included additional examples of gnotobiotic organisms in (Lines 75-81).

L104-116: "The final acclimation temperature (25°C) reflected the temperature of the subsequent experimental phases." it is unclear why the temperature needs to be 25C for the experimental phases.

R: We chose 25°C as the target temperature for the acclimation and transition phases because it reflects the average temperature of the Venice Lagoon between June and September. The Venice lagoon is one of the most important farming areas worldwide for the species and is close to our invertebrate facilities and a routine sampling site for our experiments on bivalves. By ensuring the efficacy of our protocol at this temperature, we aimed to establish its applicability in the field, beyond the laboratory setting. As mentioned in the introduction, bivalves, particularly clams, can be used as sentinel organisms to assess the effects of environmental changes such as temperature increases. In this context, 25°C serves as an optimal baseline temperature that can later be increased to study the effects of thermal stress. Ensuring that microbiome transplantation was effective at this temperature was critical to the application of the protocol to various experiments, including those related to heat stress. We have expanded the Discussion section adding the rationale behind this (Lines 414-421).

L113: Is there any bacteria present in the food used in the experiment?

R: During the development and optimisation of the protocol, we have also plated the commercial food to assess bacterial abundance. The food was plated on rich microbiological media and the population size was found to be <10 CFU/ml. Importantly, the commercial diet was always aliquoted under a hood and stored at +4°C. These details have been added to the Methods section (Lines 123-126).

L142: how were the antibiotics validated?

R: We have added the requested details in Lines 144-165.

L155: remove the extra space between morphological and differences. It also appears there might be additional instances of double spacing throughout the manuscript, please check.

R: We have replaced all double spacing throughout the manuscript.

L178-180: The reasoning behind the composition of the bacterial suspension, which includes these three specific bacteria, isn't clearly explained. I suspect it's based on the findings. Even though this is the methodology section, I would recommend the authors to incorporate a statement clarifying the origin of this selection.

R: We have expanded the 'Transplantation of the mock community' section to include additional methodological details on the selection of bacterial strains for transplantation (Lines 196-216).

L225-231: the 16S rRNA sequencing analysis paragraph lacks of details. The authors should include all the details linked to data analysis as well as provide the scripts.

R: We have expanded the Methods section related to 16S rRNA sequencing (Lines 251-263).

Not a single statistical test was realized. I strongly recommend the authors to conduct statistical analysis on their results.

R: We apologise for the confusion. The main results for which we tested statistical significance focused primarily on the efficacy of transplantation, particularly in terms of comparing the abundance of transplanted bacterial strains over time. We performed MaAsLin2 analysis, which uses generalized linear and mixed models, to assess the significant association between bacterial genera and time points. The results obtained are shown in Fig. 4F. We acknowledge that this was not clear and have expanded this section for clarity. In addition: i) we have included ANOVA tests to compare the abundances of the same bacterial strains over time and added a supplementary table to present the results (Lines 352-353, Table S3) and ii) reported statistical results for comparison between alpha diversity metrics (Fig. S2).

The authors should provide the repository information where the sequencing data has been stored and can be accessed. This is crucial for the transparency and reproducibility of the research.

R: We have added the data availability section including the accession number to the sequencing data (Line 439).

The transplantation experiment's objective is not clearly defined in the manuscript. I suggest that the authors elaborate on this in the methods section, and also provide context in both the introduction and discussion sections. This will help readers understand its relevance and importance to the overall study.

R: As mentioned above, we have expanded different sections throughout the manuscript to provide greater clarity on the rationale behind the transplantation experiment (Lines 94-97, 173-184, 400-413, 431-436).

The authors affirm that they successfully developed a gnotobiotic clam despite the detection of bacteria. I propose that in the discussion, the authors refer to previous studies with analogous results to reinforce their findings. My intention is not to imply that the clams are not gnotobiotic, but rather to preempt any potential skepticism on the part of the reader.

R: We appreciate the reviewer's comment and agree that the expansion of this section in the Discussion was necessary. It provided an opportunity to explore and address additional aspects of our proposed methodology. In response, we have included analogous examples in both the Introduction and Discussion sections to further illustrate these points (Lines 400-408).

L372-377: very interesting.

Re: Spectrum01189-24R1 (Generating gnotobiotic bivalves: a new method on Manila clam (*Ruditapes philippinarum*))

Dear Dr. Maria Elena Martino:

Thank you for the privilege of reviewing your work. Below you will find my comments, instructions from the Spectrum editorial office, and the reviewer comments.

Revision Guidelines

Sincerely,
Ruth Ann Luna
Editor
Microbiology Spectrum

Reviewer #3 (Comments for the Author):

This is a review of the manuscript
Generating gnotobiotic bivalves: a new method on Manila clam (*Ruditapes philippinarum*)
by Gallo et al. Submitted to Microbiology Spectrum

The authors present a method for creating gnotobiotic clams from the species *Ruditapes philippinarum*. The method apparently involves dunking adult clams in an antibiotic solution. I was surprised this method appeared to work, as other methods for

gnotobiotic generation often focus on sterilizing eggs and sperm and then growing the gnotobiotic adults. I am not aware of other organisms where giving antibiotics to adults removed all of the microorganisms. Therefore, I have looked at the methods that the authors used to verify the clams methods with some care.

They found that the gnotobiotic adult clams had no colony forming bacteria when they plated clam homogenate on a non-selective Marine Agar. They also performed amplicon sequencing and found the gnotobiotic bacteria appeared to be dominated by one species *Endozoicomonas elysicola*.

This paper has already gone through one round of review, and the authors have responded to those reviews. Nevertheless, I have been added as a second round reviewer. I recall not being super pleased when my papers have gone out to new reviewers, so I went into this with a goal of not doing too much nit-picking. However, I have two major concerns that I'd like the authors to address.

Generally, I find their two major claims to be not fully supported by the evidence.

First, was the claim that the clams were essentially gnotobiotic. Yes, they had no colony forming bacteria, but as the authors themselves realize, not all bacteria are colony forming. They could have actually determined the abundance of bacteria on the clams if they had used quantitative PCR or microscopy methods which are not limited to showing bacteria that can grow on agar.

The second claim is the contention that the larvae do have *Endozoicomonas elysicola*. I think these could be laboratory contaminants, rather than present in the hosts. First, since there was no quantification done so we don't know if the larvae had any microbes or not. Secondly, the authors skipped running negative controls. Negative controls would usually take the form of the authors running their DNA extraction process but with no sample, and then running that all the way through sequencing and analysis. Previous groups that have run negative controls almost always see background laboratory contaminants of some sort when doing amplicon sequencing (see references below for examples). Studies lacking laboratory controls have been shown to falsely describe the microbiome of systems in which bacteria were not present.

My first preference would be for the authors to quantify bacteria in a plate free way, and to run the negative controls. I understand that this would be difficult to do given the stage of the study. Barring this, a description of these caveats, and a discussion of what still can be gained by their presence would suffice for me.

I note that in the results, there is some endpoint PCR of *Endozoicomonas*, specifically. I didn't see how this factored into the analysis, but maybe its a legit way of determining whether it is present in the gnotobiotic clams, especially if a negative control was performed (which I did not see described).

I do find their contention that they created a clam dominated by an established mock community to be reasonable.

Beyond these objections, I found the paper to be well written and well structured. I agree that there is value in creating gnotobiotic bivalves as they enable controlled experiments into microbiota, and this paper does indeed move towards such a goal. I looked over their responses to the other reviewers and found them to be thorough.

The figures were clear and easy to follow, though I have a few minor suggestions.

Figure 2B had a lot of colors and that some of these were different to distinguish from each other. I especially struggled as I am red-green color deficient. One option would be to lump more genera into the "Other" category and to just focus on the 10-12 most abundant genera.

In Figure 4, it might help to increase the text size in many of these figures. They are a bit small and hard to read, without zooming in.

References

Tettamanti Boshier FA, Srinivasan S, Lopez A, Hoffman NG, Proll S, Fredricks DN, Schiffer JT. 2020. Complementing 16S rRNA Gene Amplicon Sequencing with Total Bacterial Load To Infer Absolute Species Concentrations in the Vaginal Microbiome. *mSystems* 5:10.1128/msystems.00777-19.

Díaz S, Escobar JS, Avila FW. 2021. Identification and Removal of Potential Contaminants in 16S rRNA Gene Sequence Data Sets from Low-Microbial-Biomass Samples: an Example from Mosquito Tissues. *mSphere* 6:10.1128/msphere.00506-21. 2.

Ducarmon QR, Hornung BVH, Geelen AR, Kuijper EJ, Zwitter RD. 2020. Toward Standards in Clinical Microbiota Studies: Comparison of Three DNA Extraction Methods and Two Bioinformatic Pipelines. *mSystems* 5:10.1128/msystems.00547-19.

Reviewer #3 (Comments for the Author):

This is a review of the manuscript

Generating gnotobiotic bivalves: a new method on Manila clam (*Ruditapes philippinarum*)
by Gallo et al. Submitted to Microbiology Spectrum

The authors present a method for creating gnotobiotic clams from the species *Ruditapes philippinarum*. The method apparently involves dunking adult clams in an antibiotic solution. I was surprised this method appeared to work, as other methods for gnotobiotic generation often focus on sterilizing eggs and sperm and then growing the gnotobiotic adults. I am not aware of other organisms where giving antibiotics to adults removed all of the microorganisms. Therefore, I have looked at the methods that the authors used to verify the clams methods with some care.

R: We thank the reviewer for their comment on this point. Indeed, to our knowledge, there is currently no method available that reliably renders adult individuals germ-free. In both molluscs and insects, existing protocols typically target the egg stage, such as embryo dechoriation using bleach or ethanol solutions, which are widely recognized as the most efficient and straightforward approaches. However, our goal was to develop a protocol that could also be applied in settings where working with larvae is not feasible, and where a faster or more acute intervention is required. We appreciate the reviewer's careful and constructive evaluation of our manuscript.

They found that the gnotobiotic adult clams had no colony forming bacteria when they plated clam homogenate on a non-selective Marine Agar. They also performed amplicon sequencing and found the gnotobiotic bacteria appeared to be dominated by one species *Endozoicomonas elysicola*.

This paper has already gone through one round of review, and the authors have responded to those reviews. Nevertheless, I have been added as a second round reviewer. I recall not being super pleased when my papers have gone out to new reviewers, so I went into this with a goal of not doing too much nit-picking. However, I have two major concerns that I'd like the authors to address.

Generally, I find their two major claims to be not fully supported by the evidence.

First, was the claim that the clams were essentially gnotobiotic. Yes, they had no colony forming bacteria, but as the authors themselves realize, not all bacteria are colony forming. They could have actually determined the abundance of bacteria on the clams if they had used quantitative PCR or microscopy methods which are not limited to showing bacteria that can grow on agar.

R: We thank the reviewer for their comment. However, we believe there may be a misunderstanding regarding the terminology used to describe the two main groups of organisms generated through our protocol.

The reviewer states:

"First, was the claim that the clams were essentially gnotobiotic. Yes, they had no colony-forming bacteria, but as the authors themselves realize, not all bacteria are colony forming."

Based on this statement, we respectfully suggest that the reviewer may be confusing "gnotobiotic" with "germ-free" or "microbiome-depleted" particularly in reference to the first paragraph of the Results section (titled "Generation of germ-free animals").

By definition, gnotobiotic animals are "animals (typically mice or zebrafish) in which normal host microbiota has been replaced by a defined set of microbes" (Basic M, Bleich A. 2019, doi:10.1177/0023677219836715; Morgan XC, Segata N, Huttenhower C. 2013, doi:10.1016/j.tig.2012.09.005). While we acknowledge that gnotobiotic has also been used as a "broad term encompassing axenic, germfree, and defined flora/fauna-associated animals (Luckey T.D. Academic Press; New York: 1963. Germfree Life and Gnotobiology), this usage is less common in current literature.

In our study, we distinguish between germ-free (or microbiome-depleted) animals (i.e., those treated with antibiotics to reduce bacterial load) and gnotobiotic animals (i.e., those colonized with a defined microbial community) (Lines 15, 98, 206-209, 422-424). As described in the first paragraph of the Results section (which we believe the reviewer is referring to), we initially tested the generation of germ-free clams using our antibiotic-based protocol. We verified this through both culture-based methods (to assess the absence of colony-forming units) and 16S rRNA amplicon sequencing (to evaluate overall bacterial richness and diversity). Although a few bacterial taxa remained detectable via sequencing, the overall bacterial diversity was significantly reduced in antibiotic-treated individuals (Fig. 2B). Subsequently, we generated gnotobiotic animals (second paragraph of the Results section) by transplanting a defined mock community into these microbiome-depleted hosts. In

accordance with the standard definition of gnotobiology, we confirmed the successful establishment of this defined microbial set.

To avoid further confusion, we have added a definition of gnotobiotic individuals to the manuscript (lines 57-59), clarifying our use of the term.

Given the reviewer's statement: "I do find their contention that they created a clam dominated by an established mock community to be reasonable" we believe our conclusions (i.e., that we generated both microbiome-depleted and gnotobiotic animals) are supported by the data presented.

The second claim is the contention that the larvae do have *Endozoicomonas elysicola*. I think these could be laboratory contaminants, rather than present in the hosts. First, since there was no quantification done so we don't know if the larvae had any microbes or not. Secondly, the authors skipped running negative controls. Negative controls would usually take the form of the authors running their DNA extraction process but with no sample, and then running that all the way through sequencing and analysis. Previous groups that have run negative controls almost always see background laboratory contaminants of some sort when doing amplicon sequencing (see references below for examples). Studies lacking laboratory controls have been shown to falsely describe the microbiome of systems in which bacteria were not present.

My first preference would be for the authors to quantify bacteria in a plate free way, and to run the negative controls. I understand that this would be difficult to do given the stage of the study. Barring this, a description of these caveats, and a discussion of what still can be gained by their presence would suffice for me.

I note that in the results, there is some endpoint PCR of *Endozoicomonas*, specifically. I didn't see how this factored into the analysis, but maybe its a legit way of determining whether it is present in the gnotobiotic clams, especially if a negative control was performed (which I did not see described).

I do find their contention that they created a clam dominated by an established mock community to be reasonable.

R: We thank the reviewer for this comment. We agree that laboratory contaminants are a common concern, particularly in studies utilizing 16S amplicon sequencing. However, we believe this is not the case for our findings, for two main reasons:

- Endozoicomonas elysicola is a well-documented symbiont of bivalves and other marine organisms (doi.org/10.1038/srep40579; doi.org/10.1007/s00253-016-7777-0), as noted in the manuscript (Lines 379-382). Given its ecological association with marine hosts, we expect its presence in our treated samples to reflect a resident microbial population rather than laboratory contamination.*
- As the reviewer noted, we performed a targeted end-point PCR on treated clams (including those found to be dominated by Endozoicomonas through 16S sequencing) using both universal 16S primers and Endozoicomonas-specific primers. This was conducted to resolve the taxonomic identity at the species level (as reported in Lines 238–248 and 307–309). The amplified sequences was identified as Endozoicomonas elysicola. Importantly, we included negative controls for both the 16S amplification and the Endozoicomonas-specific PCR. While the original manuscript reported the Sanger sequencing results (File S2), we acknowledge that we had not included the corresponding gel images, which also showed the negative controls. We have now added these results as Fig. S3, which show that all negative controls - both for the general 16S and the Endozoicomonas-specific PCRs - produced no detectable amplification.*

We hope this additional information addresses the reviewer's concern and clarifies how the end-point PCR data were used in our analysis in relation to their comment: "There is some endpoint PCR of Endozoicomonas, specifically. I didn't see how this factored into the analysis."

Beyond these objections, I found the paper to be well written and well structured. I agree that there is value in creating gnotobiotic bivalves as they enable controlled experiments into microbota, and this paper does indeed move towards such a goal. I looked over their responses to the other reviewers and found them to be thorough.

The figures were clear and easy to follow, though I have a few minor suggestions.

Figure 2B had a lot of colors and that some of these were different to distinguish from each other. I especially

struggled as I am red-green color deficient. One option would be to lump more genera into the "Other" category and to just focus on the 10-12 most abundant genera.

R: We thank the reviewer for the suggestion. We have now adjusted the colours in Fig. 2B to ensure they are clearly visible and easily distinguishable. However, regarding the threshold for the "Other" category, we have chosen not to modify it, as doing so would affect the results of multiple analyses that rely on this threshold, not only in Fig. 2, but also in Fig. 4 and the supplementary figures. We hope that the updated colour scheme improves the clarity of the bacterial genus breakdown.

In Figure 4, it might help to increase the text size in many of these figures. They are a bit small and hard to read, without zooming in.

R: We have increased the font size in the panels of Fig. 4.

References

Tettamanti Boshier FA, Srinivasan S, Lopez A, Hoffman NG, Proll S, Fredricks DN, Schiffer JT. 2020. Complementing 16S rRNA Gene Amplicon Sequencing with Total Bacterial Load To Infer Absolute Species Concentrations in the Vaginal Microbiome. *mSystems* 5:10.1128/msystems.00777-19.

Díaz S, Escobar JS, Avila FW. 2021. Identification and Removal of Potential Contaminants in 16S rRNA Gene Sequence Data Sets from Low-Microbial-Biomass Samples: an Example from Mosquito Tissues. *mSphere* 6:10.1128/msphere.00506-21.

2.

Ducarmon QR, Hornung BVH, Geelen AR, Kuijper EJ, Zwitterink RD. 2020. Toward Standards in Clinical Microbiota Studies: Comparison of Three DNA Extraction Methods and Two Bioinformatic Pipelines. *mSystems* 5:10.1128/msystems.00547-19.

Re: Spectrum01189-24R2 (Generating gnotobiotic bivalves: a new method on Manila clam (*Ruditapes philippinarum*))

Dear Dr. Maria Elena Martino:

Your manuscript has been accepted, and I am forwarding it to the ASM production staff for publication. Your paper will first be checked to make sure all elements meet the technical requirements. ASM staff will contact you if anything needs to be revised before copyediting and production can begin. Otherwise, you will be notified when your proofs are ready to be viewed.

Sincerely,
Ruth Ann Luna
Editor
Microbiology Spectrum